# The Frequency of Primary Healthcare Contacts Preceding the Diagnosis of Lower-Extremity Arterial Disease: Do Women Consult General Practice Differently?

**DOI:** 10.3390/jcm11133666

**Published:** 2022-06-24

**Authors:** Cindy P. Porras, Martin Teraa, Michiel L. Bots, Annemarijn R. de Boer, Sanne A. E. Peters, Sander van Doorn, Robin W. M. Vernooij

**Affiliations:** 1Julius Center for Health Sciences and Primary Care, University Medical Center Utrecht, Utrecht University, 3584 CX Utrecht, The Netherlands; c.p.porrasacosta@students.uu.nl (C.P.P.); m.l.bots@umcutrecht.nl (M.L.B.); a.r.deboer-9@umcutrecht.nl (A.R.d.B.); s.a.e.peters@umcutrecht.nl (S.A.E.P.); s.vandoorn@umcutrecht.nl (S.v.D.); 2Department of Vascular Surgery, University Medical Center Utrecht, 3584 CX Utrecht, The Netherlands; m.teraa@umcutrecht.nl; 3The George Institute for Global Health, Imperial College London, London SW7 2BX, UK; 4Department of Nephrology and Hypertension, University Medical Center Utrecht, 3584 CX Utrecht, The Netherlands

**Keywords:** gender differences, lower-extremity artery disease, primary care, general practitioner

## Abstract

*Background*. Women with lower-extremity arterial disease (LEAD) are often underdiagnosed, present themselves with more advanced disease at diagnosis, and fare worse than men. *Objective*. To investigate to what extent potential gender differences exist in the frequency and reasons for general practitioner (GP) consultation six months prior to the diagnosis of LEAD, as potential indicators of diagnostic delay. *Methods*. Individuals older than 18 years diagnosed with LEAD, sampled from the Julius General Practitioner’s Network (JGPN), were included and compared with a reference population, matched (1:2.6 ratio) in terms of age, sex, and general practice. We applied a zero-inflated negative binomial (ZINB) regression model. *Results*. The study population comprised 4044 patients with LEAD (43.5% women) and 10,486 subjects in the reference population (46.3% women). In the LEAD cohort, the number of GP contacts was 2.70 (95% CI: 2.42, 3.02) in women and 2.54 (2.29, 2.82) in men. In the reference cohort, 1.77 (95% CI: 1.62, 1.94) in women and 1.63 (95% CI: 1.50, 1.78) in men. In the LEAD cohort, 21.9% of GP contacts occurred one month prior to diagnosis. In both cohorts and both sexes, the most common cause of consultation during the last month before the index date was cardiovascular problems. *Conclusions*. Six months preceding the initial diagnosis of LEAD, patients visit the GP more often than a similar population without LEAD, regardless of gender. Reported gender differences in the severity of LEAD at diagnosis do not seem to be explained by a delay in presentation to the GP.

## 1. Introduction

Primary healthcare aims to protect health and treat diseases, illnesses, and injuries that can be alleviated through cost-effective and affordable interventions and programs. In the Netherlands, all inhabitants are obliged to register with a general practitioner (GP), who acts as a “gatekeeper” before contacting medical specialists [1]. GPs have the responsibility to control public healthcare costs by limiting the number of (unnecessary) referrals to specialists [2]. Recent studies find that the number of GP visits increases prior to hospitalization for an acute event, such as heart failure, myocardial infarction, and pneumonia [3,4].

Lower-extremity arterial disease (LEAD)—also known as lower-limb peripheral artery disease—is the third most common clinical manifestation of atherosclerosis after coronary artery disease and stroke. It affects over 236 million people worldwide (52.23% women) [5] and is associated with a very high risk for major adverse cardiovascular (MACE) and limb events (MALE) [6]. Previous studies have shown that women with LEAD present symptoms differently, are often underdiagnosed or not diagnosed expeditiously, and have a worse prognosis for life and limb [7,8,9,10]. Egorova et al. found that women are consistently less likely to be hospitalized for LEAD but are more prone than men to be admitted urgently. As a result, these findings could indicate that women are hospitalized with more advanced-stage disease, requiring emergent rather than elective medical treatment.

The GP is the first person to intervene at the early symptoms of LEAD [11,12]. Nonetheless, it remains unclear whether differences in severity, delays in diagnosis, and differences in prognosis between women and men might be caused, potentially due to delays, in the early stages at the GP level. There is a lack of evidence concerning discrepancies, in terms of quantity and timing, in GP visits between women and men, as a potential indicator for delays in referral. Therefore, this study aims to investigate whether differences exist between women and men in the frequency of GP contact prior to diagnosis of LEAD. Furthermore, it pursues to determine whether and when there is an increase in the number of GP consultations preceding the diagnosis.

## 2. Methods

### 2.1. Data Source

For this study, we extracted data from the Julius General Practitioner’s Network (JGPN) [13]. This ongoing dynamic database contains information of approximately 370,000 individuals registered with the involved GP centers (n = 78). The current composition of the patient database and the geographical distribution of the participating practices in urban and (semi)rural regions make the JGPN population representative of the Dutch population. Gender, mean age, and age distribution of patients are comparable to the Dutch population (47.9 vs. 49.5% males, 39.5 vs. 41.3 years). The number of participating GPs is higher than the Dutch average (60 vs. 44%), as is the number of group practices in JGPN (76 vs. 33% national average) [13].

All visits are registered according to a systematic format with information on symptoms, signs, diagnostic test results, diagnosis, and treatment of the patient, including prescription of medication and referral to hospital specialists. Diagnoses are entered following the International Classification of Primary Care (ICPC) coding, hospital referrals are coded according to referred specialism, and prescribed medication is registered using the anatomical therapeutic coding (ATC) system [13].

### 2.2. Study Population

We sampled individuals ≥18 years for two cohorts. (1) The LEAD cohort included all women and men diagnosed with LEAD for the first time, defined as ICPC codes K92 (other diseases of the peripheral arteries) or K92.01 (intermittent claudication), registered by a GP between January 2013 and February 2020. Date of LEAD diagnosis was considered the “index date”. (2) A reference population (age-, sex-, and GP-matched (1:2.6 ratio)) from the JGPN registry to compare potential sex differences in primary healthcare contact in general. Each person in the reference group served only one time as a reference.

### 2.3. Data Extraction

Information on sex, age, medical history, and the number of GP contacts was collected. Age was defined as the age at the index date and stratified as follows: <50, 50–69, 70–84, ≥85. For the medical history at baseline and number of GP contacts, we used ICPC codes. History of hypertension was defined as patients with codes K86 and K87; diabetes, T90, T90.01, T90.02; hyperlipidemia, T93, T93.01, T93.02, T93.03, T93.04; renal impairment, U99.01; vascular disease, K89 and K90; rheumatic disease, L88, L88.01, L88.02; heart disease, K75 and K76; musculoskeletal problems, L14, L15, L18, L19, L28, L90; and tobacco abuse, ICPC code P17.

### 2.4. Variables of Interest

The primary outcome of this study was the number of GP contacts; other healthcare professional contacts were not considered (i.e., nursing contact). We defined the outcome as the number of GP contacts up to six months prior to the index date by using ICPC codes recorded in their medical records. We counted each ICPC code as one contact moment. When the same ICPC code is used for multiple GP contacts, we have counted these each as unique GP contacts. The index date, i.e., time of LEAD diagnosis, was not counted as a GP contact. We classified the reason for GP consultation into six groups according to body systems, as generally is done in general practice; category A: general symptoms (e.g., fever, pain general), K: cardiovascular, L: musculoskeletal, P: psychological, S: skin, T: endocrine/metabolic, and U: urological symptoms. Factors expected to be associated with the number of GP contacts in this study were: age, history of diabetes mellitus, hypertension, hyperlipidemia, musculoskeletal problems, vascular disease, myocardial infarction, and smoking abuse [14,15].

### 2.5. Statistical Analyses

Continuous variables were presented as mean and standard deviation (SD) or as the median and interquartile range (IQR) depending on the distribution. Categorical variables were presented as absolute numbers and proportions.

Count data are assumed to have a Poisson distribution and are often analyzed using the Poisson regression model. The Poisson model is a member of the Generalized Linear Models (GLM), which assumes that mean and variance are equal. Nevertheless, in some count data, the conditional variance exceeds the conditional mean (overdispersion) and an excess of zeros (i.e., for our study, no visit to the GP) exists [16]. To account for overdispersion and excess of zeros, negative binomial regression (NB) and zero-inflated Poisson (ZIP) are used, respectively [17]. When data exhibit both problems, zero-inflated negative binomial regression (ZINB) accounts for the excess of zeros and the heterogeneity in the positive outcome. The ZINB model has two parts: an NB regression part that examines how frequent the outcome occurs (count part), and a zero-inflated part (ZI) that predicts the odds ratio (OR) of not presenting the outcome (logit part) [18]. We performed a ZINB model separately for the LEAD and reference cohort.

Because comorbidities such as hypertension, diabetes mellitus, or history of cardiovascular problems might increase the number of GP visits, ref. [15] we ran interactions models between sex and these covariates to explore potential differences between women and men with different comorbidities in frequency of GP contact. Continuous covariate (i.e., age) was mean centered in all analyses, and all tests were two-sided; *p*-value less than 0.05 was considered statistically significant. All analyses were performed using RStudio (2020, Integrated Development for R. RStudio, Inc., Boston, MA, USA) [19].

## 3. Results

In the JGPN, we identified 4044 LEAD patients and sampled 10,486 subjects for the reference cohort. Women represented 43.5% of the LEAD cohort and 46.3% of the reference cohort, and their mean age was higher than that of men (69.2 (SD 13.7) vs. 67.5 (SD 11.6) in the LEAD group and 67.0 (SD 14.2) vs. 65.2 (SD 12.2) years in the reference group, respectively). At baseline, in both cohorts, hypertension, rheumatic disease, and musculoskeletal problems were more common in women, while diabetes mellitus and myocardial infarction were more common in men (Table 1). Although the prevalence of these conditions was higher in those with LEAD compared to the reference population, differences between women and men were in the same direction for both cohorts.

### 3.1. GP Contacts Six Months Preceding the Index Date

In the LEAD cohort, 69.5% of the patients and 42.6% of the reference cohort had at least one GP contact in the six months preceding the index date (69.1% of women and 69.9% of men vs. 45.5% of women and 40.2% of men in the LEAD and reference cohort, respectively). The median number of GP contacts in the LEAD cohort was 2 (IQR: 6) for women and 2 (IQR: 6) for men. The reference cohort had a lower median number of GP contacts, 0 (IQR: 3) for women and 0 (IQR: 2) for men. These data exhibited significant variability and dispersion. The number of GP contacts ranged from 0–79 in women and 0–66 in men in the LEAD group and from 0–35 in women and 0–66 in men in the reference group (see Appendix A).

In the LEAD cohort, 3525 (21.9%) GP visits occurred one month prior to their diagnosis. During this period, women and men had the same median number of GP contacts, 2 (IQR: 2; *p* = 0.87). In the reference group, 3036 (16.2%) of the contacts occurred one month before the index date and the median number of the GP contacts for both women and men in the reference group was 1 (IQR: 1; *p* = 0.95).

For both women and men in the LEAD and reference cohort, the most common cause of consultation was cardiovascular problems, followed by endocrine/metabolic and musculoskeletal disorders. When comparing the two cohorts, the LEAD cohort had more visits due to musculoskeletal complaints than the reference group, 15% and 13% versus 12% and 8% for women and men, respectively (Figure 1a,b).

Within the LEAD cohort, the most common reason for GP visits due to musculoskeletal complaints were leg/thigh symptoms (72.3% in women and 70% in men). The reference cohort consulted more frequently due to knee symptoms (26.6% in women and 31.4% in men).

### 3.2. ZINB Model

In the LEAD cohort, the count part of the model (NB) shows that the number of GP contacts for women was 2.70 (95% CI: 2.42, 3.02). In men, this was a factor of 0.94 lower (95% CI: 0.87, 1.01); men had 2.54 (95% CI: 2.29, 2.82) contacts with the GP. The number of GP contacts was a factor of 1.77 (95% CI: 1.65, 1.91) times higher in patients with diabetes, 1.20 (95% CI: 1.10, 1.30) times higher in those with hypertension, and 1.08 (95% CI: 1.01, 1.17) for musculoskeletal symptoms.

The zero-inflated (ZI) part of the model indicates that within this cohort, the odds ratio (OR) of having zero GP contact (as compared to more than one contact) was a factor 0.94 (95% CI: 0.70, 1.26) lower in men. The OR in women was 2.70 (95% CI: 1.97, 3.68) and 2.52 (1.90; 3.34) in men. Except for rheumatic diseases and sex (men), all the other covariates included in the model decreased the ORs of having zero visits to the GP. Table 2 shows the ZINB regression model for the LEAD patients.

In the reference cohort, the number of visits to the GP for women was 1.77 (95% CI: 1.62, 1.94). In men, this was a factor 0.92 (95% CI: 0.87, 0.98) lower; thus, men had 1.63 (95% CI: 1.50, 1.78) contacts with the GP before the index date. The covariates included in the regression model incremented the rate for the number of GP visits for the reference cohort.

The OR of having no GP contact in women of the reference cohort was 6.96 (95% CI: 5.80, 8.36), lower compared with men, in which the OR of zero contact was a factor of 1.16 higher (95% CI: 0.97, 1.38); so, men had an OR 8.06 (95% CI: 6.80, 9.57) of having no GP contacts (Table 2).

Interactions between sex and covariates in the LEAD cohort were not statistically significant. In the reference cohort, interactions between sex and history of myocardial infarction and sex and hypertension decreased the number of GP contacts. None of the interactions in the ZI part of the model were statistically significant (Appendix A).

## 4. Discussion

Primary healthcare plays a critical role in the diagnosis and treatment path of patients with LEAD. GPs are the first to intervene at the earliest signs of the disease. Therefore, questions about differences in the pattern of consultations between women and men are relevant and helpful to understand whether delays in diagnosis in women could be partially explained by differences in the number of GP contacts.

This study found that women and men did not differ in the frequency of GP contacts six months prior to LEAD diagnosis. An increase in GP visits was observed one month prior to diagnosis; however, this was seen in both women and men. Similarly, the reasons for GP contact did not appear to differ between them.

In the year 2020, Dutch inhabitants had on average 5.1 contacts with the GP (95% CI: 4.8, 5.3) per person. Women had a mean of 6.2 (95% CI: 5.7, 6.6) GP contacts and men 4.0 (95% CI: 3.7, 4.3) [20]. These differences, albeit obtained over a 12-month period and for all ages, are consistent with our findings; in the reference population, matched to the LEAD cohort, during the six months previous to the index date, an average of 1.94 (3.39) visits to the GP for women occurred and 1.67 (3.32) for men.

Some studies have shown changes in the pattern of GP visits a short time before referral to a hospital due to a critical diagnosis, such as myocardial infarction or cardiac arrest [3,4,21]. This pattern was also observed in our study, in which the number of GP visits increased in the month before the index date; however, this was not different between women and men. Women had more GP contacts the month before the diagnosis, but there were no statistically significant differences between sexes. These results follow other studies that show that women and men with common morbidities have similar consultation patterns [22,23].

A considerable part of the GP contacts occurred one month prior to the index date and was labeled as a “disease of a musculoskeletal nature”. In patients with a history of atherosclerotic disease or cardiovascular risk factors (smoking, diabetes mellitus, overweight, high blood pressure, and increasing age), musculoskeletal complaints could be potential prodromal symptoms or atypical leg symptoms [24]. Unfortunately, we had no data from the free-text in the clinical notes written during the GP visit. Further studies, preferably using text-mining techniques, are needed to identify which lower-limb-related musculoskeletal symptomatology could be considered as a potential prodromal symptom. This could help GPs to identify LEAD at an early stage and potentially lead to clues about avoiding possible delays in diagnosis and referral to specialist care.

Women with LEAD have a higher frequency to be asymptomatic [25] or present with either rest pain or atypical leg symptoms; [10] these factors could contribute to a delay in diagnosis and a worse prognosis. However, it was unclear whether this phenomenon could also be partially explained by differences in the number of GP contacts prior to the diagnosis of LEAD and by delays in referral to a hospital specialist. This study provides evidence that the number of GP visits up to six months before the diagnosis of LEAD is higher than that in a matched reference cohort. Nevertheless, differences between women and men were not observed. This evidence indicates that the pattern of GP consultation was not gender-dependent; therefore, it is unlikely that the worse prognosis in women with LEAD is due to a delay in diagnosis by GPs.

### Strength and Limitations

The strengths of this study are its large sample size, its representativeness of the general Dutch population, and its reflection of routine clinical care. In addition, the use of state-of-the-art statistical models to solve analytical problems related to overdispersion and zero excess allowed for valid estimates.

We also acknowledge some limitations within this study. First, data were not collected primarily for research purposes. Therefore, its quality depends on the correct recording of the information by the GP. Second, we defined LEAD as patients with diagnosis codes K92 and K92.01 for the first time. The ICPC-1 code K92 refers to other diseases of the peripheral arteries, but within it, there are three subcategories: K92.01 (intermittent claudication), K92.02 (Raynaud’s syndrome), and K92.03 (Buerger’s disease). We classified all patients within this cohort as LEAD. However, we are uncertain how many patients classified as other diseases of the peripheral arteries (K92) have intermittent claudication (k92.01). Because we have no means of evaluating this, we cannot rule out misclassification.

Finally, we did not have information about socioeconomic status, education, and ethnicity; these factors might influence the frequency of access to the healthcare services as described by Gerritsen et al. who found a higher frequency of GP contact in women from Morocco and the Netherlands Antilles than in men coming from the same places [26]. Despite these limitations, this study is based on information that reflects routine clinical care. Therefore, we believe that it is unlikely that the limitations did influence the results significantly.

## 5. Conclusions

Patients with LEAD consulted the GP more frequently compared to the matched reference cohort, but no differences in the number of healthcare contacts between women and men were observed. Most of the consultations take place in the month before the diagnosis. Although the reasons for consultation in the last month preceding the index date were very similar, there was an increase in the number of GP visits due to musculoskeletal problems in patients with LEAD compared to the reference population. Further studies are needed to determine whether and which musculoskeletal problems could be potential early predictors of LEAD.

## Figures and Tables

**Figure 1 jcm-11-03666-f001:**
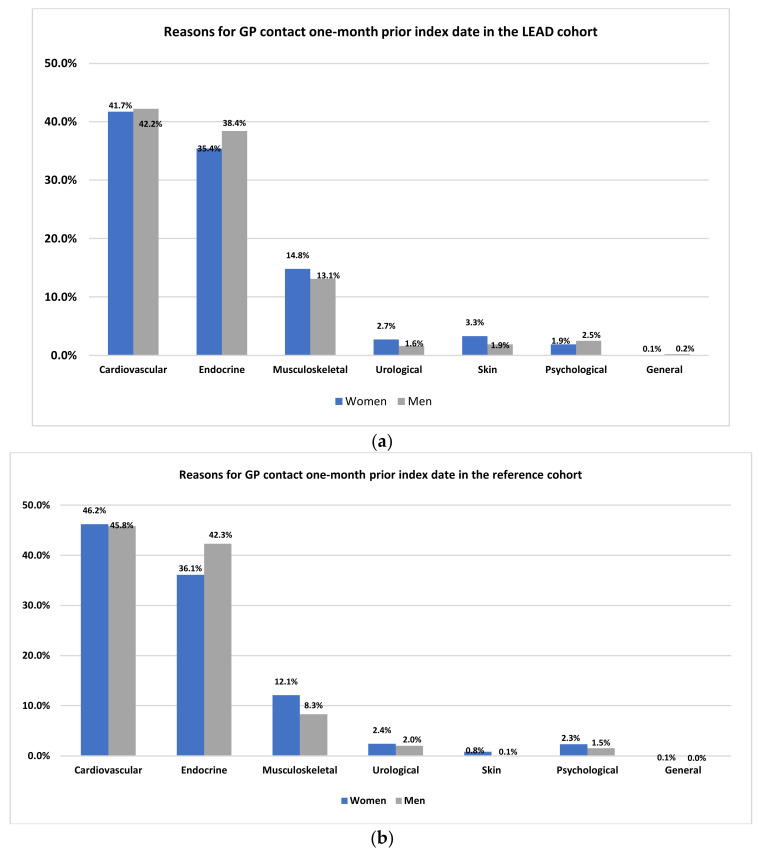
(**a**) Reasons for GP contact one month prior to index date in the LEAD group. LEAD: lower-extremity arterial dis-ease. (**b**) Reasons for GP contact one month prior to index date in the reference group.

**Table 1 jcm-11-03666-t001:** Baseline characteristics.

	LEAD Cohort ^1^			Reference Cohort		
	Women	Men	*p* Value	Women	Men	*p* Value
N	1761	2283		4851	5635	
Age (mean (SD))	69.23 (13.74)	67.55 (11.67)	<0.001	67.08 (14.25)	65.22 (12.24)	<0.001
Age group			<0.001			<0.001
<50 years (%)	159 (9.0)	159 (7.0)		561 (11.6)	589 (10.5)	
≥50 <70 years (%)	677 (38.4)	1107 (48.5)		2095 (43.2)	2990 (53.1)	
≥70 <85 years (%)	736 (41.8)	895 (39.2)		1730 (35.7)	1826 (32.4)	
≥85 years (%)	189 (10.7)	122 (5.3)		465 (9.6)	230 (4.1)	
Hypertension (%)	1111 (63.1)	1308 (57.3)	<0.001	2163 (44.6)	2193 (38.9)	<0.001
Diabetes mellitus (%)	494 (28.1)	823 (36.0)	<0.001	815 (16.8)	1035 (18.4)	0.038
Hyperlipidemia (%)	487 (27.7)	624 (27.3)	0.848	856 (17.6)	888 (15.8)	0.010
Renal impairment (%)	262 (14.9)	307 (13.4)	0.211	367 (7.6)	335 (5.9)	0.001
Rheumatic disease (%)	105 (6.0)	67 (2.9)	<0.001	193 (4.0)	139 (2.5)	<0.001
Vascular disease ^2^ (%)	225 (12.8)	307 (13.4)	0.563	310 (6.4)	405 (7.2)	0.115
MI ^3^ (%)	182 (10.3)	450 (19.7)	<0.001	202 (4.2)	500 (8.9)	<0.001
Musculoskeletal (%)	1094 (62.1)	1180 (51.7)	<0.001	2525 (52.1)	2421 (43.0)	<0.001
Tobacco abuse ^4^ (%)	523 (29.7)	670 (29.3)	0.835	467 (9.6)	577 (10.2)	0.311

N: number of patients; SD: standard deviation; LEAD: lower-extremity arterial disease; *p* value: is reflecting the differences between women and men in the specific cohort; ^1^ LEAD patients were defined as patient with ICPC codes K92 (other diseases of the peripheral arteries) or K92.01 (intermittent claudication); ^2^ vascular disease is defined as ICPC code K89 (Transient ischemic attack) and K90 (Stroke); ^3^ history of myocardial infarction; ^4^ history of tobacco abuse was defined as patients with ICPC code P17 at baseline.

**Table 2 jcm-11-03666-t002:** ZINB regression coefficients for the number of healthcare contacts in LEAD and reference cohorts.

Predictor	LEAD Cohort				Reference Cohort			
	Negative Binomial Model ^1^ (Count Model)		Zero-Inflated Model ^2^ (Logit Model)		Negative Binomial Model ^1^ (Count Model)		Zero-Inflated Model ^2^ (Logit Model)	
	Exp (β) *	CI	Exp (β) **	CI	Exp (β) *	CI	Exp (β) **	CI
**Intercept ^+^**	**2.70**	**2.42–3.02**	**2.70**	**1.97–3.68**	**1.77**	**1.62–1.94**	**6.96**	**5.80–8.36**
Sex (men)	0.94	0.87–1.01	0.94	0.70–1.26	0.92	0.87–0.98	1.16	0.97–1.38
Diabetes	1.77	1.65–1.91	0.04	0.01–0.11	2.01	1.88–2.14	0.01	0.00–0.03
Hypertension	1.20	1.10–1.30	0.11	0.07–0.17	1.31	1.22–1.40	0.06	0.05–0.08
Hyperlipidemia	1.08	1.00–1.16	0.35	0.22–0.58	1.09	1.02–1.16	0.23	0.17–0.32
Musculoskeletal	1.08	1.01–1.17	0.39	0.29–0.52	1.08	1.01–1.15	0.34	0.28–0.42
Rheumatic disease	1.09	0.92–1.29	0.62	0.25–1.50	1.25	1.10–1.43	0.17	0.09–0.33
Vascular disease ^3^	1.17	1.07–1.29	0.14	0.04–0.45	1.20	1.09–1.32	0.18	0.09–0.33
MI ^4^	1.21	1.11–1.32	0.10	0.04–0.26	1.22	1.11–1.34	0.04	0.01–0.12
Tobacco abuse ^5^	1.22	1.13–1.32	0.66	0.48–0.92	1.19	1.09–1.31	0.37	0.27–0.50
Age ^6^	1.01	1.00–1.01	0.98	0.97–0.99	1.00	1.00–1.01	0.96	0.95–0.97

CI: 95% confidence interval; ^+^ the intercept refers to a woman with mean age, the other exponent betas for the different predictor should be interpreted as factors; * exponent beta in the negative binomial part of the model is interpreted as a count; ** exponent beta in the zero-inflated (logit model) is interpreted as an odd ratio; ^1^ coefficients for the count part of the model are interpreted as predicted number of healthcare contacts; ^2^ the logistic part of the model predicts non-occurrence of healthcare contact; ^3^ vascular disease is defined as ICPC codes K89 (Transient ischemic attack) and K90 (Stroke); ^4^ history of myocardial infarction; ^5^ history of tobacco abuse was defined as patients with ICPC code P17 at baseline; ^6^ age was mean centered for all analyses.

## Data Availability

Data supporting this study are not publicly available. To access to this information please contact our research coordinator Michiel L. Bots: m.l.bots@umcutrecht.nl.

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
