# Peer review of "The Frequency of Primary Healthcare Contacts Preceding the Diagnosis of Lower-Extremity Arterial Disease: Do Women Consult General Practice Differently?"

_jcm, 2022, doi:10.3390/jcm11133666_

Round 1
Reviewer 1 Report
The study aims at reporting the number of visits to the GP prior to the diagnosis of lower extremity artery disease (LEAD).
The hypothesis ois of interest but I believe that two important issue must be solved.
1/ there is something weird and difficult to understand in the manuscript and I needed time to figure out whether authors diagnosed LEAD or PAD (including upper limb and cervical). As far as I understood this was LEAD (both symptomatic or asymptomatic). If my assumption is correct, it is mandatory that the authors change their PAD expression to LEAD, otherwise the report of approximately 1000 ICPC codes 89 becomes nonsense. If not then these +/-1000 patients should be removed.
2/ I do believe that the authors should analyze separately the subjects for wom LEAD was diagnosed for lower limb symptoms from asymptomatic LEAD, for which the LEAD diagnosis may have been based on an ABI calculation alone. Further I feel quite uncomfortable with table 2 which mixes muscle symptoms (L14, L18, L19 and L28) and joint or articular symptoms (L15, L88, L90)… clearly the global number of “musculoskeletal” contacts are similar between the cohort group and the reference cohort but it is not acceptable from my point of view to mix skeletal and muscle pain.
Author Response
Dear reviewer,
Please find attached our answers to your comments.

Reviewer 2 Report
This is a very interesting report, that partly contradicts the common issue of women being treated later than men, with worse outcome.
-outcome of any PAD treatment is obviously not reported, but would be interesting in a future publication.
-my single remark is that the registry (the investigated cohort) should be described more in detail - is the registry completely covering the population or is there a risk that certain groups are not found - based on social factors etc. ?
Author Response
Dear reviewer,
Please find attached the answer to your comments.
